# Testosterone Deficiency and Nutritional Parameters as Predictors of All-Cause Mortality among Male Dialysis Patients

**DOI:** 10.3390/nu14214461

**Published:** 2022-10-24

**Authors:** Ksymena Leśniak, Aleksandra Rymarz, Maria Sobol, Jolanta Dymus, Agnieszka Woźniak-Kosek, Stanisław Niemczyk

**Affiliations:** 1Department of Internal Diseases, Nephrology and Dialysis, Military Institute of Medicine, 04-141 Warsaw, Poland; 2Department of Biophysics, Physiology and Pathophysiology, Medical University of Warsaw, 02-004 Warsaw, Poland; 3Department of Laboratory Diagnostics, Military Institute of Medicine, 04-141 Warsaw, Poland

**Keywords:** testosterone, hormonal disorders, protein-energy wasting (PEW), chronic kidney disease

## Abstract

Background: Chronic kidney disease (CKD) is associated with an accelerated risk of cardiovascular mortality. Hormonal and metabolic disorders in CKD may constitute novel risk factors. Our objective was to characterize and evaluate prognostic implications of circulating sex steroids and selected nutritional parameters in patients at different stages of CKD. Methods: Studied groups were composed of 78 men: 31 on hemodialysis (HD), 17 on peritoneal dialysis (PD), 30 with CKD stage G3-G4. Total testosterone (TT), dehydroepiandrosterone sulphate (DHEA-S), androstenedione, luteinizing hormone (LH), prolactin (PRL), and biochemical parameters were measured; Free testosterone (FT) was calculated. Results: The lowest TT and FT were observed in HD, the highest- in CKD (*p* = 0.006 for TT, *p* = 0.005 for FT). TT positively correlated with total cholesterol in HD (*p* = 0.012), FT negatively correlated with BMI in CKD (*p* = 0.023). During the 12 months, 9 patients died (5 in the HD, 4 in the PD group). The deceased group had significantly lower concentrations of albumin (*p* = 0.006) and prealbumin (*p* = 0.001), and a significantly higher concentration of androstenedione (*p* = 0.019) than the surviving group. In the group of men on dialysis, a serum TT concentration <2.55 ng/mL (Q1-first quartile) was associated with a 3.7-fold higher risk of death, although statistical significance was not achieved (*p* = 0.198). After analysis of the ROC curves, the FT level was the best prognostic marker in HD (AUC = 0.788; 95% CI: 0.581–0.996; *p* = 0.006) Conclusions: Total and free testosterone levels were lower in the HD group than in the CKD group. The nutritional status undoubtedly affects the survival of dialysis patients but also the concentrations of testosterone significantly contributes to further worsening the prognosis.

## 1. Introduction

Patients with chronic kidney disease have a high risk of general and cardiovascular disease (CVD) mortality, and this risk increases along with the progression of kidney failure [1,2]. This fact prompts nephrologists to search for prognostic markers of mortality in this group of patients. Since this excessive risk of death cannot be completely explained by the presence and impact of classical risk factors [3], recent studies have focused on the role of non-classical risk factors, including endocrine disorders and malnutrition in patients with CKD.

Hypogonadism is the most common gonadal defect in men with CKD, and its prevalence increases with the progression of renal failure–it affects approximately 50–75% of hemodialysis patients [4,5]. Despite the broad literature on hemodialysis patients, hardly any data concerning patients undergoing peritoneal dialysis is available [6,7,8].

Hypogonadism is a group of clinical symptoms resulting from testosterone deficiency (low concentrations of total/free testosterone) explained by multifactorial causes [9,10]. Testosterone deficiency in men is associated not only with sexual dysfunction but also with an unfavorable cardiometabolic profile, both in the general population [11,12,13] and patients with chronic kidney disease [14,15,16,17,18].

Over the years, the results of many studies conducted on different populations of dialysis patients have indicated the relationship between testosterone deficiency and a higher risk of mortality, including death from cardiovascular causes [5,6,19,20,21]. However, not all studies have demonstrated that testosterone is an independent predictor of mortality [22,23,24].

There are also reports implying the association between adrenal androgens, especially dehydroepiandrosterone sulfate, and mortality; as well as the incidence of cardiovascular disease in men. However, the results of epidemiological studies are contradictory [25,26], and so far, only a few studies have been performed on the population of patients with CKD. The studies conducted on the Asian population revealed that low DHEA-S levels were a significant predictor of all-cause and cardiovascular mortality in men undergoing hemodialysis [27,28].

Nutritional state parameters cannot be omitted in the assessment of the factors that affect mortality in men with CKD. Protein-energy wasting (PEW, according to International Society on Renal Nutrition and Metabolism 2007) occurring in approximately 20% of CKD patients correlates with the severity of the renal failure. The highest percentage of patients with signs of malnutrition is found among dialysis patients [29]. This state is associated with a higher incidence of complications, hospitalizations, and mortality in this population [30,31]. Testosterone deficiency, which is an anabolic hormone, contributes to the development of sarcopenia, malnutrition, and frailty syndrome in CKD [32,33,34].

Our objective was to characterize and evaluate prognostic implications of circulating sex steroids and selected nutritional parameters in patients at different stages of CKD.

## 2. Materials and Methods

This is an observational, prospective study conducted in a population of patients who were under medical care provided by the Department of Internal Diseases, Nephrology and Dialysis Therapy and the Nephrology Outpatient Clinic of the Military Institute of Medicine in Warsaw.

Seventy eight male patients aged between 41–89 years: 31 on hemodialysis, 17 on peritoneal dialysis, 30 with chronic kidney disease stage G3-G4, according to the Kidney Disease: Improving Global Outcomes (KDIGO) definition (eGFR < 60 mL/min/1.73 m^2^ and ≥15 mL/min/1.73 m^2^), were included in the study. Patients were studied in metabolic steady state.

Dialysis patients had spent at least 3 months on dialysis therapy. Patients on HD received hemodialysis for 12 h per week, none was on an on-line technique. Patients on PD were receiving continuous ambulatory peritoneal dialysis (CAPD *n* = 16) or automatic peritoneal dialysis (APD *n* = 1). The adequacy of dialysis in HD patients was assessed monthly (measured by single pool Kt/V). In PD patients, adequacy of dialysis was assessed monthly (weekly Kt/V, and the residual renal function was assessed by urea and creatinine clearance).

The exclusion criteria for the study were as follows: lack of Informed consent to participate in the study; active neoplastic disease (patients at the stage of oncological diagnosis or treatment); hormone therapy for endocrine diseases (including androgen supplementation); severe state of secondary hyperparathyroidism; severe clinical condition, including severe inflammation; recurrent dialysis peritonitis and cirrhosis of the liver.

The study protocol was accepted by the Military Institute of Medicine Bioethics Committee (approval number 50/WIM/2016). The study was performed in accordance with the Declaration of Helsinki.

The tests were performed twice: at the study entry and after 12 months. In the first stage of the study, a detailed medical history was collected from each patient, a physical examination was performed, and medical records were analyzed. The concentration of albumin, prealbumin, total cholesterol, leptin, sex hormone binding globulin (SHBG), dehydroepiandrosterone sulphate, androstenedione, total testosterone, luteinizing hormone, prolactin, and parathyroid hormone (PTH) was measured in the patients’ blood serum.

Additionally, free testosterone levels were calculated from total testosterone, SHBG, and albumin using The International Society for the Study of the Aging Male (ISSAM) calculator available at http://www.issam.ch (accessed on 1 September 2022).

eGFR was achieved using the Modification of Diet in Renal Disease (MDRD) formula [35].

In each patient, Body Mass Index (BMI) was also determined.

After 12 months of observation, selected elements of the medical history and physical examination were repeated, and control laboratory tests were performed.

The group of men on peritoneal dialysis was assessed only at baseline, because the restrictions introduced in connection with the COVID-19 pandemic made it impossible to fully assess patients after 12 months.

Blood samples for testing were collected in the morning between 7.00–11.00, in patients on hemodialysis before the dialysis session. Serum was separated and kept frozen at −70 °C, if not analyzed immediately.

For the laboratory determination of examined parameters, the following methods were used: total testosterone, DHEA-S, LH, SHBG, PRL, PTH–electrochemiluminescence method (Roche Elecsys analyser, Mannheim, Germany). Androstenedione, leptin- ELISAkit (DRG, Marburg, Germany). Prealbumin–nephelometric method (BN II Siemens). Creatinine, albumin and total cholesterol tests were performed by routine methods at the Department of Laboratory Diagnostics of the Military Institute of Medicine and Laboratory Department of Clinic of Endocrinology and Internal Diseases of the Medical University of Warsaw.

Total testosterone deficiency was considered if the serum level was <2.88 ng/mL; 10 nmol/L (normal values were 2.8–8.2 ng/mL).

Free testosterone deficiency was considered if calculated free testosterone concentration was <50 pg/mL; <0.17 nmol/L [9].

Cardiovascular diseases included [coronary artery disease, myocardial infarction, heart failure, cerebral ischemic stroke, central nervous system (CNS) hemorrhage, symptomatic peripheral arterial disease of the lower extremities].

All-cause mortality, as well as the date and etiology of death, were collected after reviewing the available information. In addition to all-cause mortality, cardiovascular death (CV death) was defined as death due to sudden cardiac death, coronary heart disease, peripheral arterial disease of the lower extremities, and stroke [23].

### Statistical Analysis

A statistical analysis was conducted using the Statistica1 3.3 package (StatSoft Poland, Dell Statistica Partner). The quantitative data were summarized using descriptive statistics (mean ± SD, median and range). Binary data were summarized as percentages. Nominal variables were analyzed by the Chi-square test. The distribution of each variable was checked for consistency with normal distribution (Shapiro-Wilk test). To assess the differences between groups, the t-test or the nonparametric Mann–Whitney test was used depending on the fulfillment of the assumption of normal distribution. Differences between dependent variables were analyzed using the paired t-test or Wilcoxon’s test, respectively. The nonparametric Kruscall-Wallis test was used to evaluate differences between more than two groups. Because of multiple comparisons, the Bonferroni correction was included. The Pearson or the Spearman rank correlation test were used to assess correlations of androgens with other variables. The receiver-operating characteristic (ROC) analyses was performed to identify the best cut-off point value of serum androgens that could help recognize worse treatment outcomes in the dialysis groups. The area under the curve (AUC) was calculated for each considered parameter. The results were considered as statistically significant if the *p* value was less than 0.05 or 0.017 in case of multiple comparisons.

## 3. Results

### 3.1. General Characteristics

In total, 78 male patients were included in the study. The mean age of the CKD group was 67.6 ± 9.2 years, HD group 61.4 ± 10.0 years, and PD group 59.2 ± 12.2 years. There were no statistically significant differences in age between the groups (*p* = 0.082). The mean creatinine concentration in the CKD group was 2.3 ± 0.8 mg/dL, and the mean estimated glomerular filtration rate (eGFR) was 32.6 ± 10.6 (mL/min/1.73 m^2^). The mean duration of hemodialysis and peritoneal dialysis was 2.75 ± 2.69 years and 1.7 ± 1.6 years, respectively.

The characteristics of the study groups, including comorbidities and their frequency, are included in Table 1.

The characteristics of the study groups including selected nutritional parameters are included in Table 2.

The characteristics of the study groups including hormonal profile are included in Table 3.

The highest percentage of patients with total testosterone deficiency (TT < 2.88 ng/mL) was found in the HD group (54%) vs. (23%) in the PD group and (23%) in the CKD group. Similarly, the highest percentage of patients with free calculated testosterone deficiency (FT < 50 pg/mL) was observed in the HD group (51%) vs. (23%) in the PD group and (10%) in the CKD group.

Serum DHEA-S deficiency (<70.2 ug/dL) was found in all groups (from 23% of men in the PD group, 26% in the CKD group to 32% in the HD group). Serum androstenedione deficiency (<0.91 ng/mL) was found in all groups as well (from 29% of men in the PD group and 36% in the CKD group, up to 45% in the HD group).

Statistically significant differences between the groups were observed for the following hormones: total testosterone, free testosterone, and PRL (Table 3).

A statistically significant higher level of total testosterone (*p* = 0.006) and free testosterone (*p* = 0.005) was found in the CKD group compared with the HD group. There were no statistically significant differences between the HD and PD groups for serum concentration of TT, but for FT the difference was at the border of significance (*p* = 0.025).

A statistically significant lower level of PRL (*p <* 0.001) was found in the CKD group compared with the HD group.

After 12 months of observation, there was no statistically significant difference in TT, FT, or androstenedione serum concentration in the CKD or HD groups, while for DHEA-S, there was a statistically significant decrease in the concentration from the value (median 109 µg/dL) to the value (median 100 µg/dL) in the CKD group (*p* = 0.014).

### 3.2. Correlation Analysis

Total testosterone positively correlated with total cholesterol (in the HD group r = 0.444, *p* = 0.012). Free testosterone negatively correlated with BMI (in the CKD group r = −0.414, *p* = 0.023). DHEA-S negatively correlated with total cholesterol (in the CKD group r = −0.374, *p* = 0.042), BMI (in the HD group r = −0.373, *p* = 0.046), and leptin (in the PD group r = −0.556, *p* = 0.020 and in the HD group r = −0.444, *p* = 0.014). Androstenedione negatively correlated with leptin in the HD group (r = −0.404, *p* = 0.024). Leptin positively correlated with BMI in the HD group (r = 0.853, *p <* 0.001) and CKD group (r = 0.827, *p <* 0.001), (Figure 1).

However, no statistically significant correlation between age and total testosterone was found (*p* = 0.204 in CKD; *p* = 0.250 in HD; *p* = 0.636 in PD group). On the other hand, DHEA-S negatively correlated with age in the CKD group (r = −0.522, *p* = 0.003).

There were no statistically significant correlations between PTH and androgen concentrations or selected nutritional parameters in any of the groups.

### 3.3. Implications and Outcomes

In the group of 78 men, 9 individuals died during the 12-month follow up period. In the HD group, 5 patients died and 4 underwent transplantation. In the PD group, 4 patients died and 5 underwent transplantation. In the CKD group, all patients were alive at the end of the observation period.

The main cause of death was cardiovascular disease (8/9 deceased men); only 1 person died from sepsis (1/9 deceased men). All deceased men had a history of CVD, and 8/9 of the dead men had diabetes. The mean age of the deceased patients was 62 ± 11 years vs. 60 ± 11 years for those who survived (*p* = 0.728).

The 9 deceased men had statistically higher concentrations of androstenedione in the serum (*p* = 0.019) and statistically lower concentrations of albumin (*p* = 0.006) and prealbumin (*p* = 0.001) compared with the remaining 39 men who survived.

In this study, no statistically significant differences were found between the group of men who died and the rest of participants in total serum testosterone concentration, free serum testosterone concentration, or serum DHEA-S concentration (Table 4).

For assessing the probability of survival, the group of male dialysis patients (HD + PD groups) was divided according to the concentration of total testosterone in the blood serum. Twelve men were below the lower Q1 quartile (TT < 2.55 ng/mL), 12 men were above the upper Q3 quartile (TT > 3.82 ng/mL), and 24 men were between the Q1 and Q3 quartiles.

In men undergoing dialysis (hemodialysis or peritoneal dialysis) with serum total testosterone levels <2.55 ng/mL, the risk of death was 3.7 times higher compared with patients with testosterone concentrations >3.82 ng/mL. However, this association failed to reach statistical significance (*p* = 0.198) (Figure 2).

The analysis of the ROC curves demonstrated the importance of the free testosterone level in the serum as a prognostic marker in men undergoing hemodialysis (*p* = 0.006); (OH: AUC = 0.788; 95% CI: 0.581–0.996) (Table 5). The proposed cut-off value of free testosterone concentration for the assessment of prognosis is 54.6 pg/mL (Figure 3). It is worth noting that the concentration of androstenedione as a prognostic marker in this group of patients was at the border of statistical significance (Table 5).

The analysis of the ROC curves demonstrated the importance of the androstenedione level in the serum as a prognostic marker in men undergoing dialysis (hemodialysis or peritoneal dialysis) (*p* = 0.003); (OH: AUC = 0.761; 95% CI: 0.591–0.931). The proposed cut-off value of androstenedione serum concentration for the assessment of prognosis is 1.33 ng/mL (Figure 4).

Additionally, Kaplan-Meier analysis for all-cause mortality in men undergoing HD according to the presence of diabetes mellitus and cardiovascular diseases was performed. There was no statistically significant difference in survival for any divisions of subjects. The obtained *p* values are 0.087 for the division according to diabetes presence and 0.923 for cardiovascular disease, respectively.

## 4. Discussion

Our study assessed a wide range of hormonal disorders including androgens and their relationship with mortality in the population of male patients with CKD stage G3-G4 and those undergoing dialysis.

In this study, the groups of men on hemodialysis and peritoneal dialysis did not differ significantly in the mean concentration of androgens. As expected, the group of men with CKD had statistically higher mean concentrations of total testosterone (*p* = 0.006) and free calculated testosterone (*p* = 0.005) compared with male hemodialysis patients. Predictably, the highest percentage of testosterone deficiency (TT < 2.88 ng/mL; <10 mmol/L) was demonstrated in hemodialysis patients (54%), which is greater than in the general population but similar to the results of other studies involving hemodialysis patients [6,22,23,36]. The percentage of men with FT deficiency was lowest in the CKD group (10%) and increased in the dialysis groups (23% in peritoneal dialysis patients and as much as 51% in the HD group), which is consistent with the previous observations [14,37].

While the occurrence of hypogonadism has been well-documented in hemodialysis and CKD patients, little is known about its prevalence in the peritoneal dialysis population [6,7,8,38].

In this study, we report that the percentage of men with testosterone deficiency in the PD group was 23% (diagnosed on the basis of both total testosterone and calculated free testosterone levels). This result is more than half lower than in the group of hemodialysis patients.

The reasons for a high prevalence of hypogonadism in patients undergoing dialysis are unclear. Although some testosterone may be removed from circulation by hemodialysis, increased clearance cannot explain the low levels of testosterone [39,40]. *Cigarran* et al. previously demonstrated the difference in the prevalence of testosterone deficiency between patients undergoing different types of dialysis. In their study, in a group of 79 patients, testosterone deficiency was found in the majority of men on hemodialysis (39.5%), and only in 5.6% of patients undergoing peritoneal dialysis. The authors speculate that these differences may be associated with lower testosterone removal during peritoneal dialysis compared with hemodialysis, and/or greater protein loss during peritoneal dialysis, which may result in greater elimination of protein-bound testosterone and the increase in the free testosterone pool. Appropriate studies of hormone removal during dialysis are required to verify these hypotheses [38].

Apart from testosterone, adrenal androgens were also assessed. The mean serum concentrations of DHEA-S and androstenedione were within the normal range, while the percentage of patients with decreased DHEA-S concentration ranged from 23% in the peritoneal dialysis group and 26% in the CKD group, to 32% in the HD group. In turn, reduced serum concentration of androstenedione was observed in 29% of the PD group, 36% in the CKD group, and 45% of hemodialysis patients. The obtained results provide evidence for abnormalities in all androgen levels, especially in hemodialysis patients.

So far, alterations in circulating DHEA-S levels in end-stage renal disease have been studied only in a few small studies [41]. Kakija et al. demonstrated that serum DHEA-S levels were significantly lower (by 40–53%) in men undergoing hemodialysis compared with healthy subjects in the control group (771 ng/mL vs. 1650 ng/mL, respectively) [27]. In turn, Hsu et al. demonstrated a mean concentration of DHEA-S equal to 809.7 ng/mL in 200 men undergoing hemodialysis [28]. Similar values of DHEA-S concentrations were found in our population of hemodialysis patients (median 94.5 ug/dL = 945 ng/mL), and it was lower in this group compared with PD or CKD patients.

Importantly, we did not find any data in other studies from the last 20 years considering the assessment of disturbances in serum androstenedione concentration in the population of patients with advanced renal failure [42].

Moreover, in this study, men with advanced renal failure had elevated mean baseline serum levels of LH, SHBG and leptin, while dialysis patients also had higher levels of prolactin. These results are in agreement with the previously described endocrine disorders in CKD [43,44].

In the presented study, dialyzed males (HD + PD) were divided into groups based on total testosterone levels. At that time, the risk of death of dialyzed men on hemodialysis or peritoneal dialysis turned out to be 3.7 times higher when serum concentrations of TT were <2.55 ng/mL compared with the TT concentrations in the range >3.82 ng/mL. However, this association failed to reach statistical significance (*p* = 0.198).

The majority of studies suggest a relationship between testosterone deficiency and higher mortality in patients with chronic kidney disease [5,14,20] and dialysis patients [6,19,20,21,22,23,24,36].

The comparison of the aforementioned results with the literature data reveals that the value of a lower norm limit for total testosterone, which is associated with an increased risk of death in our study, corresponds to the results of other researchers. Carrero et al. analyzed a cohort of Swedish hemodialysis patients and demonstrated that men with total testosterone levels in the lowest range of the limit of the norm (TT < 2.33 ng/mL) had a higher total risk of death and death from cardiovascular causes (hazard ratio: HR 2.03; 95% CI 1.24–3.31; *p* = 0.004) and (HR 3.19; 95% CI 1.49–6.83; *p* = 0.008), respectively, compared with those in the highest range (TT > 3.45 ng/mL) [23].

Similarly, in a Japanese study of hemodialysis patients, all-cause mortality was significantly higher in patients with total testosterone levels in the low range (TT < 2.61 ng/mL) compared with the higher range (TT >3.95 ng/mL) with the hazard ratio of: 2.26; 95% CI: 1.21–4.23; *p* = 0.01 [36]. In turn, *Yu* et al. suggest that total testosterone levels below ~2.96 ng/mL are associated with a lower survival of male hemodialysis patients in the USA. The risk of mortality decreased with increasing testosterone concentrations, until the threshold of 4.0 ng/mL was reached. At that point, the risk was stabilized [6].

In the single Polish publication concerning the relationship between testosterone and prognosis in patients with CKD, Niemczyk et al. implied that serum free testosterone concentration might be a better predictor of survival than age [17]. Likewise, in the presented study, FT, with a cut-off value of 54.6 pg/mL determined on the basis of the ROC curve, was the best prognostic parameter among all androgens in male hemodialysis patients.

It is assumed that the use of TT concentration to diagnose hypogonadism in elderly men could lead to underdiagnosis; therefore, plasma FT assessment is recommended. In the present study, the percentage of male hemodialysis patients with hypogonadism diagnosed on the basis of total testosterone levels and calculated free testosterone did not differ significantly, but the latter parameter had a higher prognostic value than total testosterone. The value of the assessment of free or bioavailable testosterone levels has not yet been established, perhaps due to the fact that such measurements could be either difficult or unreliable, and currently they are not routinely or commercially available [45,46].

For many years, an increasing number of articles concerning the relationship between adrenal androgens, especially DHEA-S, with mortality and the occurrence of cardiovascular diseases in men with renal failure have been published. In the aforementioned publication, Kakija et al. demonstrated that low serum DHEA-S levels were a significant predictor of all-cause mortality in Japanese men undergoing hemodialysis, regardless of a history of cardiovascular disease and markers of inflammation and malnutrition [27].

Contrary to expectations, no relationship between DHEA-S concentration and mortality of dialyzed men was found in the present study (*p* = 0.384). However, it should be noted that the Japanese study was conducted on a larger cohort of patients (313 men on hemodialysis), and the follow-up period was longer and lasted 5 years [27].

Since the normal plasma DHEA-S concentration in patients with chronic kidney disease is unknown, it is difficult to determine cut-off values for DHEA-S deficiency. Based on the ROC curve, the determined cut-off point for the assessment of prognosis for DHEA-S was in our study 78 µg/dL and corresponds to the value determined by Hsu et al. Based on the results of multivariate analysis, these researchers suggest that low plasma DHEA-S levels are significantly and independently associated with overall mortality in male hemodialysis patients (HR = 2.933; *p* = 0.033), and the prognostic threshold for DHEA-S established with the use of the ROC curve is 79 ug/dL [28].

For the first time in the available literature, we present the association between an elevated concentration of androstenedione in the blood serum with the mortality in male dialysis patients.

In our study, a significantly higher serum level of androstenedione was found in the group of dialyzed men who died compared with the rest of the male group (*p* = 0.019). The interpretation of these results in the context of the available data introduces considerable difficulties. We have not found articles showing similar relationships, so more research should be conducted to verify them. Perhaps men with chronic kidney disease, especially those undergoing dialysis, develop complex disorders of testosterone metabolism.

It is worth mentioning that, in our study, nutritional status parameters (albumin, prealbumin) had prognostic value in terms of survival. In the group of 9 male dialysis patients who died, significantly lower concentrations of albumin (*p* = 0.006) and prealbumin (*p* = 0.001) were found in comparison with the remaining 39 men.

It should be underlined that the mean concentration of albumin and prealbumin determinations in the group of men who died correspond to the criteria of malnutrition (PEW) according to the latest nomenclature (Table 4). These results are in line with the current reports on the association of malnutrition with increased mortality in patients with advanced renal failure [47].

Moreover, androgen concentrations correlated with the parameters of nutritional status and leptin in the studied groups. These results may indicate a relationship between androgens and the nutritional status as well as the metabolism in men with renal insufficiency. Similar associations have been described in previous studies [5,22,23]. For example, in the large cross-sectional study involving 420 hemodialysis male patients in Turkey, an increase in total testosterone levels by 1.0 nmol/L was associated with a reduction in total mortality by 7% during 32 months of follow-up. However, this relationship was dependent on age and other risk factors, including BMI, albumin and creatinine levels. Gungor et al. reported that serum testosterone levels positively correlated with creatinine and inversely correlated with age, body mass index, and lipid levels [22].

Our results on the inverse correlation of testosterone with BMI and the positive correlation between BMI and leptin are consistent with previous studies on the relationship between obesity and hypogonadism [48,49] and are in line with the hypothesis that, in obesity, through direct action of leptin, testosterone production may be suppressed [50].

Here are some limitations of our study.

First, because this study was an observational study, we cannot confirm a causal relationship between androgens and nutritional parameter levels and the outcome.

Second, we were limited by the relatively small sample size and short follow-up period.

Third, total testosterone was assessed based on the ECLIA method, which has its limitations against the gold standard of liquid chromatography-mass spectrometry.

Fourth, we did not collect information about the clinical symptoms of hypogonadism; however, they are nonspecific and difficult to distinguish from the symptoms of CKD.

Furthermore, the monocenter source of data may limit the validity of our results, which should be confirmed in further studies.

In conclusion, it should be recognized that circulating sex steroids may have prognostic significance, as well as nutritional parameters and should be treated as research issues, which require further development and potentially alternative treatment strategies in CKD. Further studies are needed also considering the important variations of the enzyme systems in which zinc and copper play a decisive role in influencing the levels of the considered hormones.

## Figures and Tables

**Figure 1 nutrients-14-04461-f001:**
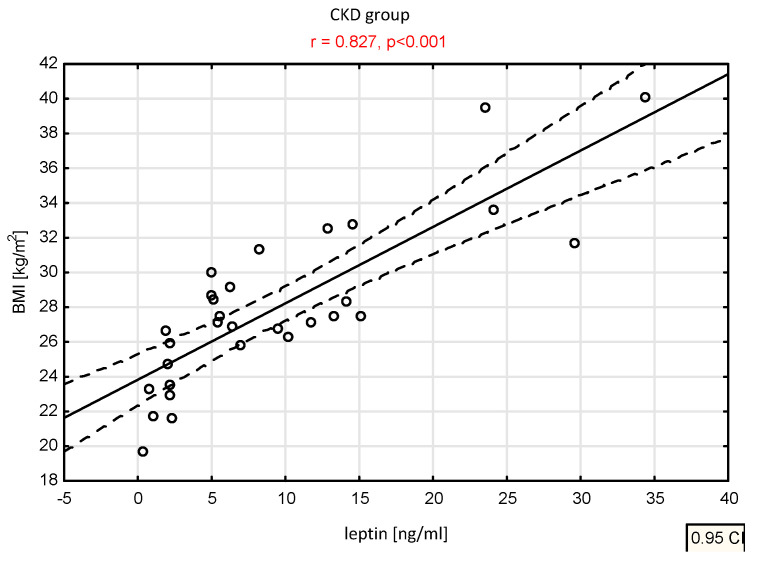
Correlation between leptin concentration and BMI in the CKD group.

**Figure 2 nutrients-14-04461-f002:**
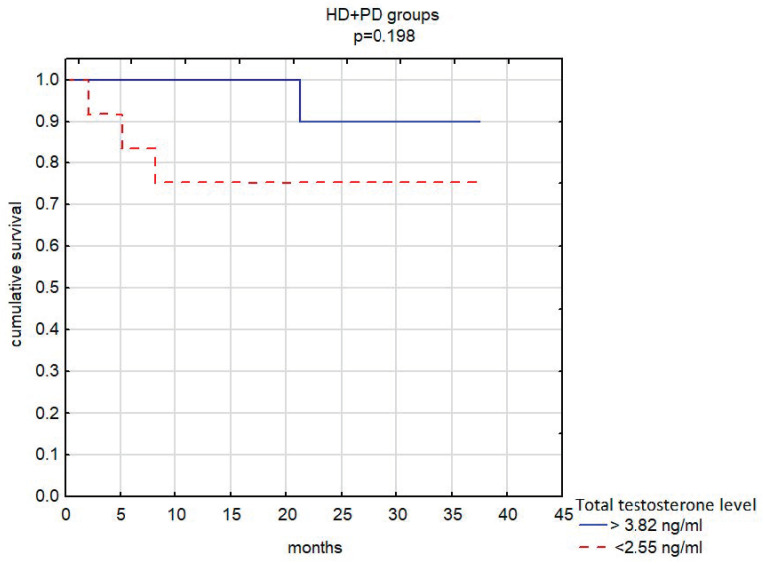
The probability of survival in the group of men on dialysis depending on the concentration of total testosterone in the blood serum.

**Figure 3 nutrients-14-04461-f003:**
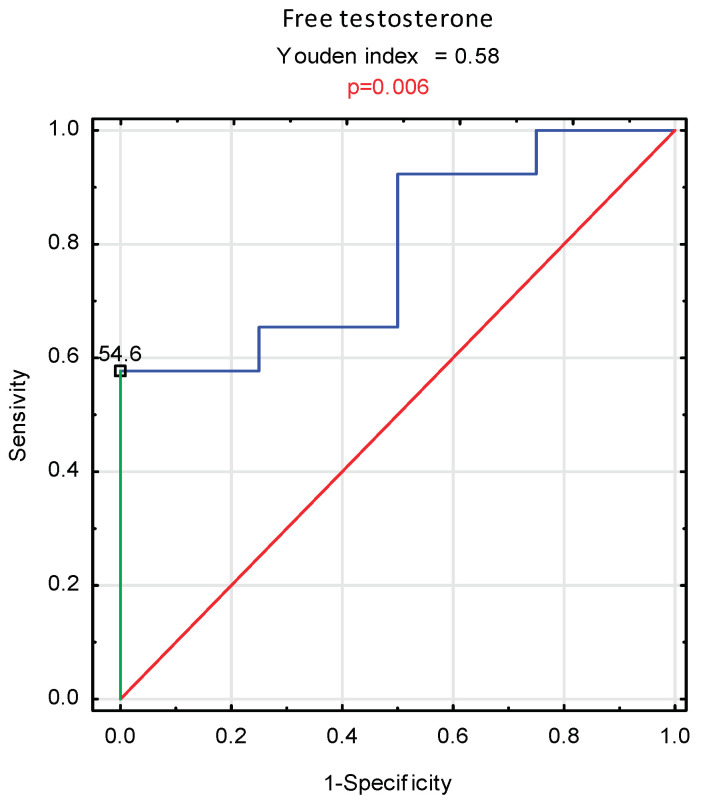
ROC analysis of free testosterone serum concentration in the prognosis prediction in men undergoing hemodialysis.

**Figure 4 nutrients-14-04461-f004:**
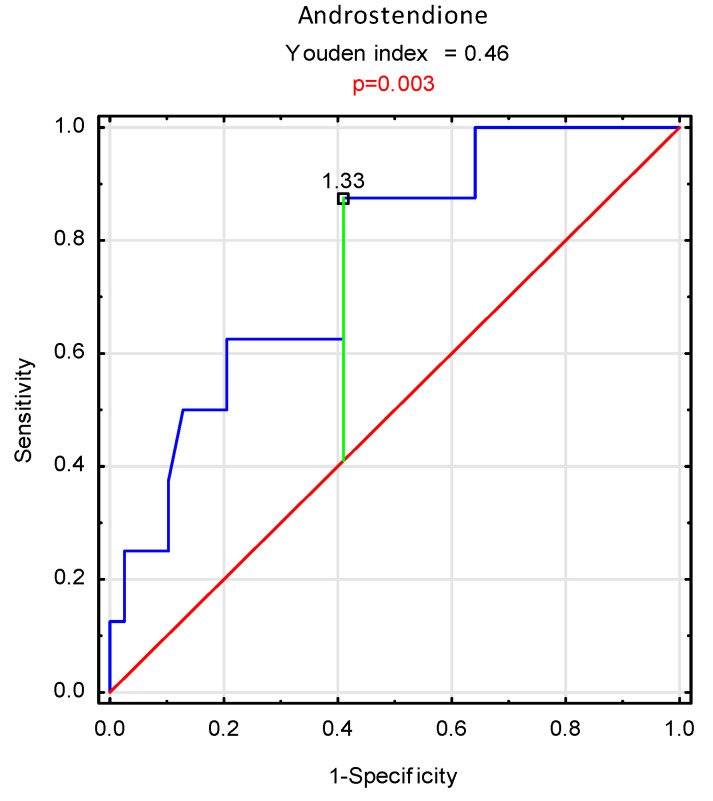
ROC analysis of androstenedione serum concentration in the prognosis prediction in men undergoing dialysis (HD + PD).

**Table 1 nutrients-14-04461-t001:** The characteristics of the studied groups of men including comorbidities.

	CKD	HD	PD	*p* *
CVD history	40.0% (12/29)	74.2% (23/31)	64.7% (11/17)	**0.031**
DM	33.3% (10/29)	61.3% (19/31)	47.0% (8/17)	0.115
Hypertension	83.3% (25/29)	100.0% (31/31)	100.0% (17/17)	0.152
Smoking	10.0% (3/29)	9.7% (3/31)	11.8% (2/17)	0.923

CVD-cardiovascular diseases, DM = diabetes mellitus. * Overall comparison in Kruskal-Wallis test. Bolded value is statistically significant.

**Table 2 nutrients-14-04461-t002:** Selected parameters of nutritional status in the study groups.

		CKD	HD	PD	*p* *
		*n* = 30	*n* = 31	*n* = 17	
BMI (kg/m^2^)	Mean ± SD	28.0 ± 4.7	28.9 ± 5.0	27.3 ± 4.2	0.521
	Median (min–max)	27.3 (19.8–40.1)	28.2 (18.9–38.8)	27.3 (18.9–53.1)	
Albumin (g/dL)	Mean ± SD	4.4 ± 0.3	4.1 ± 0.4	3.8 ± 0.4	**<0.001**
	Median (min–max)	4.5 (3.9–4.9)	4.1 (3.2–4.7)	3.9 (2.5–4.2)	
Prealbumin (mg/dL)	Mean ± SD	33.9 ± 7.5	31.6 ± 8.5	30.3 ± 7.2	0.533
	Median (min–max)	32.5 (19–56)	31 (21–54)	33 (13–40)	
Total cholesterol (mg/dL)	Mean ± SD	161.3 ± 48.0	145.8 ± 39.1	151.3 ± 33.8	0.434
	Median (min–max)	149 (96–247)	140 (90–252)	152 (81–230)	

* Overall comparison in Kruskal-Wallis test. Bolded value is statistically significant.

**Table 3 nutrients-14-04461-t003:** Assessment of the hormonal profile in the studied groups of men.

		CKD	HD	PD	*p* *
		*n* = 30	*n* = 31	*n* = 17	
Total testosterone (ng/mL)	Mean ± SD	3.8 ± 1.4	3.1 ± 1.3	3.7 ± 1.3	**0.016**
	Median (min–max)	3.9 (0.03–8.1)	2.7 (1.1–8.0)	3.6 (1.7–6.4)	
Free testosterone (pg/mL)	Mean ± SD	68.6 ± 24.3	53.8 ± 18.6	69.5 ± 23.2	**0.010**
	Median (min–max)	65.1 (0.5–123)	49.0 (24.9–105)	67.8 (40.9–133)	
DHEA-S (ug/dL)	Mean ± SD	143.5 ± 96.9	127.2 ± 88.0	181.7 ± 169.6	0.441
	Median (min–max)	109 (35–418)	94.5 (46–331)	143 (9–689)	
Androstenedione (ng/mL)	Mean ± SD	1.2 ± 0.7	1.3 ± 0.7	1.6 ± 0.8	0.173
	Median (min–max)	1.1 (0.3–3.0)	1.1 (0.5–2.9)	1.7 (0.2–3.1)	
SHBG (ug/mL)	Mean ± SD	4.6 ± 1.4	4.9 ± 3.3	4.6 ± 1.7	0.626
	Median (min–max)	4.3 (2–7)	4.2 (2.0–18.9)	4.9 (1.6–8.0)	
Leptin (ng/mL)	Mean ± SD	9.4 ± 8.7	15.4 ± 15.2	10.9 ± 8.7	0.277
	Median (min–max)	6.3 (0.3–34.4)	11.8 (0.4–72.1)	8.6 90.5–29.8)	
LH (IU/L)	Mean ± SD	11.7 ± 8.9	15.4 ± 16.5	18.5 ± 14.7	0.191
	Median (min–max)	9.1 (0.1–48.1)	9.8 (0.4–79.4)	12.8 (5.8–57.5)	
PRL (ng/mL)	Mean ± SD	10.8 ± 5.5	30.6 ± 27.5	22.4 ± 11.3	**<0.001**
	Median (min–max)	9.3 (4.2–31.2)	19.7 (9.4–132.5)	17.1 (13.3–57.0)	
PTH (pg/mL)	Mean ± SD	-	379.5 ± 319.9	401.5 ± 196.4	0.358
	Median (min–max)	-	291.3 (21.3–1461)	373.7 (152.1–788.3)	

* Overall comparison in Kruskal-Wallis test. Bolded values are statistically significant.

**Table 4 nutrients-14-04461-t004:** Comparison between the group of men who died and the rest of male dialysis patients by selected nutritional parameters and the concentration of hormones in the blood serum.

Parameter	Non Survivors			Survivors			*p*-Value *
	*n*	Mean ± SD	Median	*n*	Mean ± SD	Median	
Albumin (g/dL)	9	3.6 ± 0.5	3.9	39	4.1 ± 0.3	4.1	**0.006**
Prealbumin (mg/dL)	9	21.8 ± 8.1	22.5	39	33.1 ± 6.6	34.0	**0.001**
Total cholesterol (mg/dL)	9	137.4 ± 29.0	133.5	39	150.6 ± 38.6	148.0	0.411
BMI (kg/m^2^)	9	27.4 ± 5.0	25.8	39	28.6 ± 4.7	25.8	0.383
Total testosterone (ng/mL)	9	2.8 ± 0.9	3.1	39	3.4 ± 1.4	3.3	0.350
Free testosterone (pg/mL)	9	49.6 ± 18.1	48.0	39	61.7 ± 21.9	58.3	0.234
Androstenedione (ng/mL)	9	2.0 ± 0.8	2.1	39	1.3 ± 0.7	1.1	**0.019**
DHEA-S (ug/dL)	9	148.3 ± 93.6	125.5	39	146.9 ± 133.1	98.0	0.384
PTH (pg/mL)	9	281.8 ± 181.3	301.5	39	411.6 ± 295.0	358.9	0.250

* Overall comparison in Mann–Whitney test. Bolded values are statistically significant.

**Table 5 nutrients-14-04461-t005:** ROC analysis of androgen serum concentration in the prognosis prediction in men undergoing hemodialysis.

Parameter	Cut-Off Value	Sensitivity	Specificity	AUC	Significance-*p*
Total testosterone (ng/mL)	3.65	0.31	0.01	0.615	0.441
Free testosterone (pg/mL)	54.6	0.59	0.00	0.788	0.006
DHEA-S (ug/dL)	78.0	1	0.57	0.62	0.238
Androstenedione ng/mL	0.8	0.88	0.41	0.721	0.066

## Data Availability

Not applicable.

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
