# Peer review of "Testosterone Deficiency and Nutritional Parameters as Predictors of All-Cause Mortality among Male Dialysis Patients"

_nutrients, 2022, doi:10.3390/nu14214461_

Round 1

Reviewer 1 Report

In the manuscript “Testosterone deficiency and poor Nutritional status as a predictors of all-cause mortality among dialysis men”, the authors proposed that “dialysis men serum concentration of testosterone and nutritional parameters had prognostic value in terms of survival”. The paper has scientific merit because it brings novelty in the determination of possible hallmarks of the disease progressions however, it has some drawbacks. The work was well conducted and has scientific merit. This manuscript needs that some aspects to be elucidated. All sections need careful revision before the manuscript could be considered for publication. I do believe that the scientific outcome will significantly increase after the following revisions.

Specific comments:

11.    There are some phrases with redundancy throughout the introduction such as the expression “cannot be not completely”. Please try to clarify.

22.    In the introduction, the subject of hypogonadism was approached very lightly, which later makes it difficult to understand the results and the discussion.

33.    Throughout the manuscript, the acronyms appear once, at other times their extended form appears. please uniform.

44.    In the materials and methods section, some aspects remain to be elucidated

4.1 What is the relevance of having 16 individuals with continuous ambulatory peritoneal dialysis and having only 1 individual receiving automatic peritoneal dialysis?

4.2 The reason why the adequacy of dialysis in HD patients is done monthly and the PDs are done weekly. 

55.    In the discussion section, some points disagree with the description of the results.

5.1 In the sentence “the percentage of men with FT deficiency was the lowest in the CKD group ….”, the data demonstrates that FT deficiency was not in this group. Please rectify. 

5.2 Some p values are not the same as in the results, such as the values p=0.384 and p=0.0019.

66.    Be careful that in the discussion there are consecutive paragraphs that begin with the same word, such as “So far”, “Moreover”, “aforementioned”. Please try to conduct the discussion with fewer repetitions.  

77.    In the opinion of the reviewer, the discussion is too extended as it focuses too much on similar approaches rather than trying to explain the relevancy of the results reported.

Reviewer 2 Report

General comment

The manuscript entitled “Testosterone deficiency and poor nutritional status as predictors of all-cause mortlity among dialysis men” by Lesniak et al., aims to evaluate the role of testosterone and other circulating sex steroids in patients with different stages of CKD. Albeit the manuscript if overall well written, few corrections have to be implemented in order to improve the quality of your work and make it suitable for publication. The suggested corrections are reported below.

-          Major issues

INTRODUCTION

Regarding the role of testosterone and its effects on cardiovascular disease and CKD, see DOI: 10.3390/ijms23073535 and DOI: 10.1186/s12916-020-01594-x

State the aim of your study in a more precise manner. What do you want to report? Which is the objective of your paper?

RESULTS

The results, albeit interesting, seems to be too verbose and lengthy. This could bring the reader to lose the focus of your manuscript, worsening the overall quality of your work. Try to be objective in reporting your findings.

DISCUSSION

The discussion is overall fairly written. However, the role of leptin and the findings that you retrieved on this topic are not properly discussed. This should be improved.

Moreover, in the abstract, you said that you wanted to evaluate the role of sex hormones in different stages of CKD. However, it seems more than it is in different types of CKD treatment.

-          Minor issues

ABSTRACT

The mean age of patients reported in the background could be safely deleted or reported in the result section of the abstract.

Try to make the results clearer, avoiding too many words and using absolute number and percentages for outline differences between groups.

INTRODUCTION

Rewrite the sentence outlining the higher risk of patients in dialysis. Writing “especially those on dialysis” without giving a minimum of info is not sufficient.

MATERIALS AND METHODS

This is superfluous as it is implied that only patients who signed an informed consent would participate in the study.

TABLES

Table 1: report p values among groups.

REFERENCES

Try to use references not older than 2010, when possible.

OTHER

Improve English grammar and check typos.

Round 2

Reviewer 1 Report

the revised version has improved

Author Response

Dear Reviewer,

thank you for careful and detailed reviews of our manuscript.

Reviewer 2 Report

The authors successfully implemented the suggested comments.

Author Response

(The authors gave the same response as above.)
